ecology

defaunation, deforestation, exploitation, forest biodiversity, forest cover change, Living Planet Index

**Author for correspondence:**
Louise McRae
e-mail: Louise.Mcrae@ioz.ac.uk

†These authors contributed equally to this work.

# Below the canopy: global trends in forest vertebrate populations and their drivers

Elizabeth J. Green[1,†], Louise McRae[2,†], Robin Freeman[2], Mike B. J. Harfoot[1], Samantha L. L. Hill[1,3], William Baldwin-Cantello[4] and William D. Simonson[1]

[1]UN Environment World Conservation Monitoring Centre, 219 Huntingdon Road, Cambridge, UK
[2]Institute of Zoology, Zoological Society of London, London, UK
[3]Department of Life Sciences, Natural History Museum, Cromwell Road, London SW7 5BD, UK
[4]WWF-UK, The Living Planet Centre, Woking, UK

EJG, 0000-0002-2458-0172; LM, 0000-0003-1076-0874; MBJH, 0000-0003-2598-8652; SLLH, 0000-0003-0565-6554

Global forest assessments use forest area as an indicator of biodiversity status, which may mask below-canopy pressures driving forest biodiversity loss and 'empty forest' syndrome. The status of forest biodiversity is important not only for species conservation but also because species loss can have consequences for forest health and carbon storage. We aimed to develop a global indicator of forest specialist vertebrate populations to improve assessments of forest biodiversity status. Using the Living Planet Index methodology, we developed a weighted composite Forest Specialist Index for the period 1970–2014. We then investigated potential correlates of forest vertebrate population change. We analysed the relationship between the average rate of change of forest vertebrate populations and satellite-derived tree cover trends, as well as other pressures. On average, forest vertebrate populations declined by 53% between 1970 and 2014. We found little evidence of a consistent global effect of tree cover change on forest vertebrate populations, but a significant negative effect of exploitation threat on forest specialists. In conclusion, we found that the forest area is a poor indicator of forest biodiversity status. For forest biodiversity to recover, conservation management needs to be informed by monitoring all threats to vertebrates, including those below the canopy.

## 1. Introduction

As we arrive at the 2020 expiration of the Aichi Biodiversity Targets under the Convention on Biological Diversity (CBD), the continuing loss of biodiversity remains a seemingly intractable environmental challenge [1] with grave implications for human wellbeing and the supply of valuable ecosystem services [2]. Some 322 vertebrates have become extinct since 1500, and more than 27% of all assessed extant species are threatened with extinction [2,3]. At a global scale, the average abundance of monitored vertebrate populations declined by 60% between 1970 and 2014 [4]. With the average rate of vertebrate species loss over the last century being up to 100 times the background rate, there is little doubt that we have entered an era representing the sixth mass extinction [1].

Deforestation has been a significant driver of this worldwide biodiversity crisis. Over a century ago, most clearance was of temperate forests [5], leading to observed species extinctions [6], while in the last decades, the main deforestation frontiers and risks to biodiversity have been in the tropics [7,8]. Tropical forests are some of the most biodiverse ecosystems on Earth, harbouring over half the world's terrestrial species [9]. Yet, deforestation of tropical forests, reducing their land coverage from 12% to less than 5% [10], along with their degradation and fragmentation, have resulted from large-scale industrial and local subsistence agriculture [11] as well as logging, fires and road building

[12]. This represents a loss of important resources and habitat for humanity (between 1.2 and 1.5 billion people are directly dependent on ecosystem services provided by tropical forests [13]) as well as biodiversity, with far-reaching implications for the climate system [14] and global carbon cycle [15,16].

Land-use change is predicted to continue as a major driver of terrestrial biodiversity loss for the rest of this century [17]. In order to assess the impacts of land conversion pressures, it is crucial to develop national-to-global scale biodiversity measurements [18]. Owing to the importance of forests as habitat for many species, forest area is often employed as an indicator in global agreements and processes aimed at slowing and reversing the decline of biodiversity. Under the Strategic Plan for Biodiversity 2011–2020, for example, Aichi Target 5 focuses on halving the rate of loss of forests and other natural habitats by 2020 [19]. The suite of indicators for Sustainable Development Goal (SDG) 15 (Life on land) of the 2030 Agenda on Sustainable Development includes forest area as a proportion of total land area, and the proportion of forest and other ecosystems covered by protected areas [20]. Similarly, indicators used to monitor biodiversity conservation in the Forest Resources Assessment of the Food and Agricultural Organization (FAO) comprise area of primary forest, forest area designated for the conservation of biodiversity and forest area in legally established protected areas [21]. However, the pertinence of forest area as a relevant indicator of forest biodiversity has never been tested at a global scale. While habitat loss is the major driver of forest biodiversity loss, a focus on forest area alone risks masking other pressures on forest vertebrates that can operate below the canopy in conjunction with or independently of forest cover change. Consequently, areas with stable or increasing forest cover might be experiencing undetected declines in forest vertebrates, leading to the so-called empty forests that appear intact but have lost many of their large animals [22].

Understanding the status of forest biodiversity is important not only for species conservation but also because biodiversity loss can have consequences for forest health [12,23] and carbon stocks [24,25]. The status of the world's forests is a critical factor in the avoidance of dangerous climate change, with afforestation or reforestation being critical to many of the scenarios consistent with meeting the 1.5°C target [26]. Concurrently, the conservation of biodiversity in forests can have direct carbon benefits. Forest vertebrates, particularly large birds and primates, play an important role in forest regeneration and long-term carbon storage [27]. A loss or reduction in forest vertebrates from regions with a high proportion of large-seeded animal-dispersed tree species, such as Africa, Asia and the Neotropics, can lead to carbon losses in forests [24,25,28]. Defaunation therefore threatens the role that forests play as essential carbon stores and sinks, risking the investments made by governments and non-state actors in forests as carbon 'banks'.

Using the Living Planet Index (LPI) methodology [29,30], we aimed to develop the first global indicator of forest vertebrate specialist populations to improve assessments of forest biodiversity status. Given the decline in area of natural forest over time [31] and the link between habitat loss and biodiversity loss [32], we expected to find that forest vertebrates were in decline. We then assessed whether trends in forest vertebrate populations were related to changes in tree cover, derived from satellite-derived tree cover datasets that matched the forest vertebrate data in space and time. If tree cover were a good indicator of forest biodiversity, we would expect to find a positive relationship between forest vertebrate population change and tree cover change. We therefore tested two hypotheses:

(1) Forest vertebrates are in decline worldwide.
(2) Forest vertebrate population change is positively correlated to tree cover change.

## 2. Methods

### (a) Development of a Forest Specialists Index

The Living Planet Database (LPD) contains time-series abundance data for over 22 000 vertebrate populations including more than 4200 species across the globe, with the earliest records dating back to the 1950s (www.livingplanetindex.org). The data are collated from a range of sources, including peer-reviewed literature, grey literature, online databases and data holders. Metadata associated with each population, such as taxa, region, biome or habitat association, are also entered into the database.

The decision to develop an indicator for forest specialists as opposed to all forest species follows the approach, but not the same method of selection, of the indicators developed for European birds [33]. Given that specialists depend entirely on forests, their use in this indicator would provide a better representation of ecosystem health. We defined forest specialists using the habitat coding from the IUCN Red List [3]. Those with 'Forest' listed as one of multiple major habitats for that species were considered forest generalists, while those with only 'Forest' listed as the major habitat were considered forest specialists. This definition of specialist is narrow as the 'Forest' category from the IUCN Red List refers to natural habitat and does not include artificial habitats such as plantations. However, as the category applies to the major habitats a species occurs in, it is still possible that all or part of a population may be located in or adjacent to a plantation. The forest specialists dataset comprised 268 forest specialist species (455 populations): 135 birds, 89 mammals, 19 reptiles and 25 amphibians. See electronic supplementary material, S1 for a breakdown by realm and taxonomic class.

We followed the approach of the diversity-weighted LPI [30] to create a weighted index proportional to the species richness of each biogeographic realm and taxa in the dataset, and also to enable results to be compared with the global terrestrial LPI. In order to calculate weightings for each taxon and realm, the total number of vertebrate species from each taxonomic class and biogeographic realm that have 'Forest' listed as a habitat was taken from the IUCN Red List. Unlike for birds, mammals and amphibians, the coverage of reptile assessments in the IUCN Red List is not comprehensive so we did not have a full list of forest reptile species globally. However, the number of forest reptiles by realm was considered usable, given that the proportion of reptile species in each realm was similar to amphibians and also because spatial patterns of species richness tend to be similar among other vertebrate groups [34].

To create the subsets for the indicator, we disaggregated the data according to three taxonomic groups (mammals, birds, herptiles) by five realms (Nearctic, Palaearctic, Neotropical, Afrotropical, Indo-Pacific). Combining amphibians and reptiles into a herptile group, and Indo-Malaya, Australasia and Oceania into a single Indo-Pacific realm was a response to low data availability for these subsets. The final combinations yielded a total of 14 subsets as there were no time-series data available for Palaearctic herptiles.

The Forest Specialist Index was calculated using the R package rlpi (https://github.com/Zoological-Society-of-London/rlpi) following the approach in McRae et al. [30]. The weightings

calculated above for forest species were applied to each of the 14 subsets. In order to examine trends within these subsets of the data and by forest biome, we compared the mean and standard error of the species trends within each of the subsets. The individual species trends were available as one of the outputs of the rlpi package.

## (b) Correlates of forest vertebrate population change
### (i) Forest populations and tree cover change
While the Forest Specialist Index reflects population changes in forest specialists to more accurately reflect ecosystem health, changes in tree cover may also affect populations of forest generalists. We, therefore, selected all forest specialists and generalists that were surveyed at a specific location (defined as a discrete area such as a national park or sample area of a forest; a non-specific location comprises a larger survey area such as a province or country). For each population, the period encompassing the first and last year of survey data is subsequently referred to as the study period. Many population records do not have data available for every year of the study period. We determined annual predicted abundance values per population by fitting generalized additive models (GAMs) to the time-series population data where survey data were available for at least 6 years, and linear regressions where data were available for between 2 and 5 years, following Spooner et al. [35].

In order to assess the relationship between tree cover change and forest vertebrate populations, we required a continuous measure of tree cover spanning multiple years and at a resolution that is sensitive to the local changes that are likely to be relevant to populations. Various global datasets exist that provide continuous tree cover values for multiple years and vary in tree cover definition, spatial resolution, temporal coverage and frequency (electronic supplementary material, S2). Currently, the highest resolution global datasets (e.g. approx. 30 m) are available for a shorter temporal coverage than some datasets with a coarser resolution. Higher resolution datasets allow more fine-scale detection of changes in vegetation cover, while longer-term datasets increase the likelihood of detecting a relationship between tree cover change and population change by increasing the number of populations and years that can be analysed. We opted to run our analyses twice, once using the shorter-term (2000–2017) 30 m Landsat Global Forest Change dataset (hereafter referred to as the Hansen dataset; [36]) and once using the longer-term (1982–2016) 5.6 km MEASURES VCF5KYR dataset, which includes annual fractional tree cover and bare ground cover values (hereafter referred to as the Song dataset; [8]). In addition to fractional tree cover in 2000 and 2010 (2010 layer accessed from [37]), the Hansen dataset provides annual tree cover loss as a binary presence/absence value for 2000–2017, defined as complete stand replacement or a change from a forest to a non-forest state within a pixel. This information allows the estimation of deforestation rates, but may mask fine-scale changes within pixels such as a reduction (but not complete loss) in tree cover and assigns gradual losses that occur over multiple years to a single year.

It is important to note that, while the 30 m dataset used in these analyses comes from the Global Forest Change dataset, neither this nor the Song dataset differentiate between natural, semi-natural or non-natural forests (such as plantations). Thus, while losses (or gains) in tree cover might reflect deforestation (or regeneration) in natural forests, in plantations, this might reflect harvest (or growth) of products grown specifically for human extraction that may provide lower quality habitat for forest vertebrate populations. Systematically collected global data on tree plantations are lacking. The Global Forest Watch (GFW) Tree Plantations layer records tree plantations in a single year (2013/2014) for only seven countries [38] and is,

therefore, unsuitable for our analyses. A recently released near-global dataset on plantations by GFW [14] is also unsuitable, as the reference year is 2015. In the absence of suitable global information distinguishing natural and planted forests, we, therefore, refer to tree cover rather than forest cover whenever discussing values derived from the spatial tree cover datasets used in this analysis.

We fitted a 5 km radius around each population, based on the mean range size across all forest populations (electronic supplementary material, S3), and extracted annual tree cover area and bare ground area for 1982–2016 using the Song dataset and tree cover area in 2000 and 2010 using the Hansen dataset. We additionally extracted annual loss values for 2001–2017 from the Hansen dataset, using per-pixel tree cover in 2000 to estimate how much tree cover was lost per buffer per year. All data extraction was carried out in Google Earth Engine [39]. We plotted annual tree cover values from the Song dataset against year to visually assess temporal changes in tree cover per location. We identified substantial inter-annual fluctuations in tree cover at some locations that were unlikely to reflect true changes. To smooth these fluctuations in the Song dataset, GAMs were fitted to the annual tree cover values within each buffer to obtain annual fitted tree cover values.

We reduced the annual fitted population data to only include years that fell within 1982–2016 when analysing the effects of tree cover change with the Song dataset and 2000–2015 when analysing with the Hansen tree cover dataset. In both cases, we removed populations that no longer had greater than or equal to 2 years of data spread over at least a 5-year period (electronic supplementary material, S4 and S5). Using the annual logged values from the GAM and linear regression performed earlier, we calculated an average rate of change value per each remaining population as our response variable, following Spooner et al. [35]. Using the Song dataset, we reduced the annual fitted tree cover values to match the study period of each population, with a 1-year lag (i.e. tree cover in year $t$ matched to population data in year $t+1$). We then calculated three predictor variables from the fitted tree cover values: mean tree cover during the study period; mean bare ground cover during the study period; and the tree cover trend over the study period, taken as the year coefficient from an ordinary least-squares regression of annual fitted tree cover on year. We also calculated three predictor variables from the Hansen dataset: tree cover in 2000; the area of tree cover lost over the study period (based on loss data only); and the proportional change in tree cover between 2000 and 2010 (as these are the two years with percentage tree cover per pixel available). We removed populations with zero tree cover in all years from the analyses, leaving 1668 generalist and 175 specialist populations in the analyses using the Song dataset compared with 685 generalist and 74 specialist populations in the analyses with the Hansen dataset (see electronic supplementary material, S3 and S4 for a breakdown by realm and taxonomic class, respectively). Fewer populations were included in the analyses with the Hansen dataset because the shorter temporal period covered by the Hansen dataset (2000–2015) meant fewer populations had data overlapping that period, compared with the longer-term Song dataset (1982–2016).

In order to examine the agreement between the two tree cover datasets, we calculated tree cover change per population from 2000 to 2010 using values derived from the Song dataset and from the Hansen dataset. We then assessed the correlation between the two sets of tree cover change values for the 685 populations included in the Hansen analyses. The correlation between the two datasets was highly significant but had a low correlation coefficient (Pearson correlation coefficient = 0.171; $p < 0.001$). This is in agreement with other studies that have found discrepancies between tree cover datasets when assessing tree cover change or area [40,41].

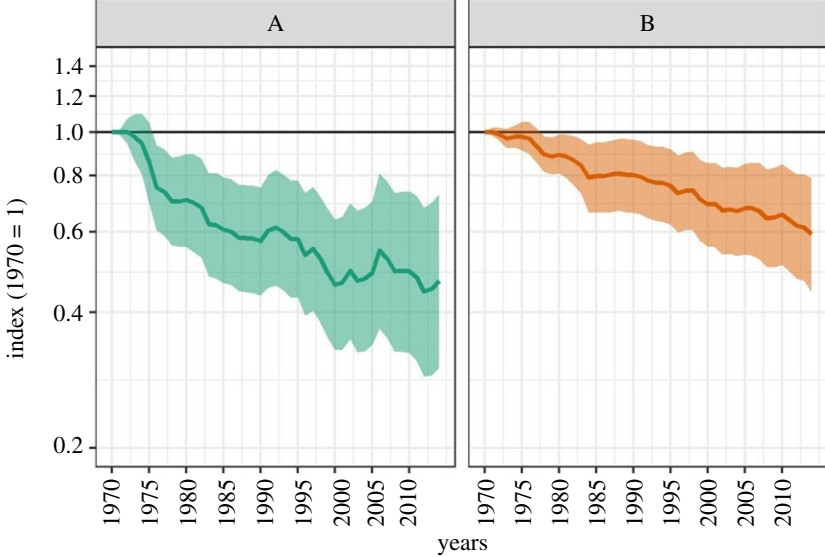

**Figure 1.** Weighted index of population change from 1970 to 2014 for (*a*) 268 forest specialist species and (*b*) 1853 terrestrial species (includes the forest specialist species). Solid line shows the weighted index values and shaded region shows the 95% confidence for the index. (Online version in colour.)

### (ii) Additional drivers of forest population change

Forest vertebrates are affected by many drivers that may occur independently of, or in conjunction with, tree cover change. We selected correlates for our analyses through a literature review and information stored in the LPD, which includes any threats specified by the source of the population data. Exploitation, including the hunting, persecution, indirect killing or collection of wild individuals for trade, is likely to be a key driver of some forest vertebrate populations [42]. We, therefore, included in our analyses a binary variable specifying whether the primary threat to the population was or was not exploitation. It is possible that body size may impact species' sensitivity to forest change [43]. To investigate this effect, we took adult body mass values per species from the Amniote [44], AmphiBIO [45] and Elton-Traits 1.0 [46] databases. Where species-level body mass information was not available, we assigned the species the mean body mass of its genus, family or order (higher taxonomic ranks used where data were unavailable for lower ranks). The body mass values were log-transformed (base 10) to normalize them. We calculated the density of roads within the study area, defined as the total length of roads within each population's 5 km buffer, using the gROADS v. 1 dataset [47]. We used the UN-Adjusted Gridded Population of the World V. 4 dataset [48] to calculate the mean human population density (HPD) within each buffer in the year 2000. Finally, we calculated the mean travel time to the nearest city or densely populated area for each buffer from the Accessibility to Cities 2015 dataset [49].

### (iii) Model structure

At some locations, multiple populations were monitored over the same period, so we chose to fit a model to the data that would take into account their non-independence. For each predictor variable, we fitted mixed effects models using the 'lme4' package [50] with the average rate of change of each population as the dependent variable, location as a random effect and the predictor as a fixed effect. We fitted separate models for each predictor variable to identify any relationships between these variables and population change, with the aim of fitting multivariate models where evidence of a relationship was found for more than one predictor variable. To determine whether a predictor variable was a significant driver of population change, we calculated Akaike's information criterion (AIC) for all models and compared them with the AIC of the null model including only a random effect of location. We considered a predictor variable

to have significantly improved the model fit if inclusion of the variable lowered the AIC by at least 2 compared with the null model (a more negative AIC indicates a better model fit; [51]).

We fitted these models to all forest populations (generalists and specialists) and additionally to forest specialist populations only. All analyses were carried out in the statistical software R v. 3.5.1 [52].

### (iv) Influential genera

We investigated whether any groups of species were having a significant influence on the models. In the absence of any groups of influential species, models iteratively excluding one group at a time would not produce substantially different model estimates. We used the 'influence.ME' package [53] to produce estimates from models that iteratively excluded the influence of each genus, where each predictor variable was fitted in a univariate mixed effects model with genus as a random effect. We used the 'sigtest' function to test whether excluding any genus changed the statistical significance of any of the predictor variables in our models. We then examined the influential genus to determine the cause and, if the genus was known to be responding to a driver other than those included in our analyses (e.g. disease, poisoning), we repeated our analyses with the genus omitted.

## 3. Results

### (a) Forest Specialist Index

The Forest Specialist Index declined by 53% between 1970 and 2014 (figure 1*a*; index value: 0.47; range 0.30–0.73). This indicates an average decline in 455 monitored populations of forest specialists at an annual rate of 1.7% per year. By comparison, the terrestrial LPI declined by 41% between 1970 and 2014 (figure 1*b*; index value: 0.59; range 0.44–0.79), representing an average decline for 5175 monitored terrestrial populations with an annual rate of 1.2% per year. The decline in the Forest Specialist Index was steepest between 1970 and 1976. The percentage of all species that had an annual declining trend was consistently between 50 and 65% during the time period except for the late 1980s, early 2000s and 2013–2014, when the proportion dropped below half (electronic supplementary material, S6). These

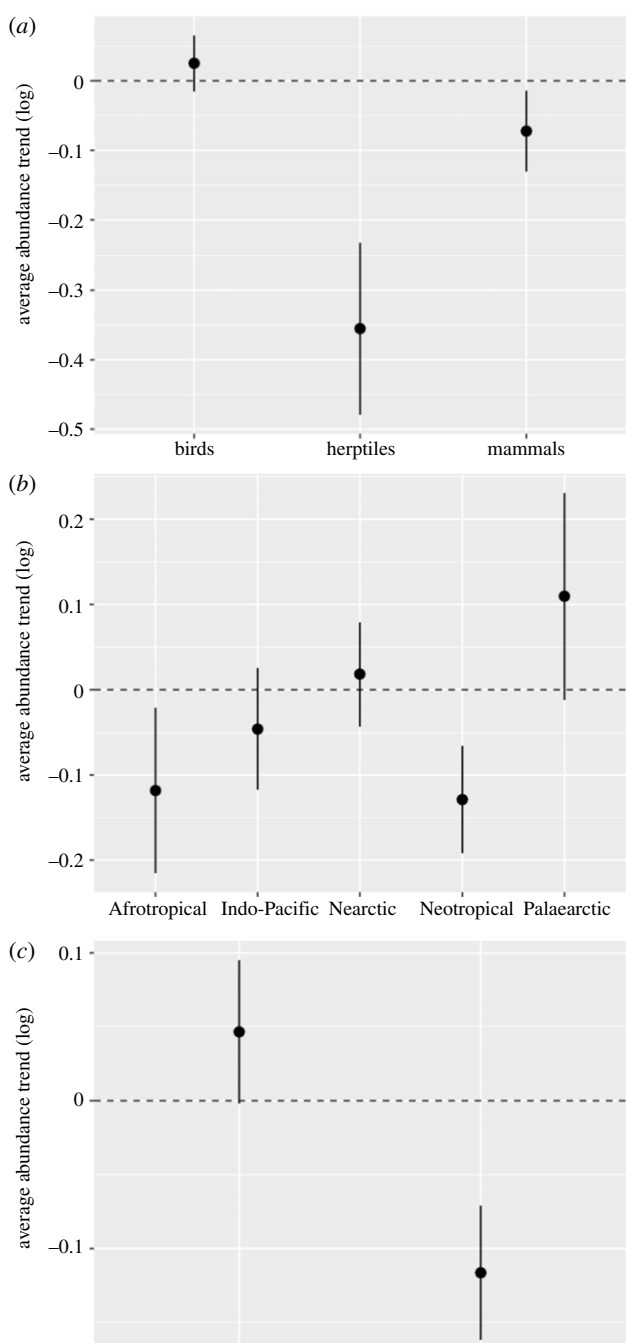

**Figure 2.** Total rate of change in forest specialist populations averaged by species, with standard error. Comparison by class (*a*), realm (*b*) and biome group (*c*). (Online version in colour.)

declining annual trends varied across the time series (electronic supplementary material, S7). The average rate of change per species was negative for tropical realms and tropical biomes and positive for temperate realms and biomes (figure 2), with no overlap between the error bars for the two biome groups. Similarly, the number of declining species trends from tropical realms and tropical forest biomes was greater than increasing (electronic supplementary material, S8), while the reverse was true of temperate realms and temperate forest biomes (electronic supplementary material, S8).

### (b) Correlates of forest vertebrate population change

We identified one genus (*Gyps*) that had a large influence on the model estimates. *Gyps* vultures are a group of generalist species that have declined severely since the 1990s because of accidental poisoning from the veterinary drug diclofenac [54], and are, therefore, a very specific case that does not reflect responses of forest populations to any of the widespread pressures we have investigated. We, therefore, excluded *Gyps* vultures from our analyses.

Mixed effects models including specialist and generalist forest populations and using the long-term Song tree cover dataset showed no evidence of a relationship between forest population change and tree cover trend (figure 3), mean tree cover, mean bare ground, exploitation, HPD, mean travel time or road density (electronic supplementary material, S9).

We found a significant negative effect of exploitation on forest specialist population change, although this was based on exploitation being the primary threat to just 12 out of 175 forest specialist populations. We found no evidence of a relationship between forest specialist population change and any other predictor variable (electronic supplementary material, S10).

Mixed effects models including forest specialists and generalists and using the Hansen tree cover dataset found no evidence of any relationships between population change and any predictor variables (electronic supplementary material, S11). We found no significant relationships between any predictors and population change when repeating the analyses using only forest specialist data (electronic supplementary material, S12).

## 4. Discussion

Our results indicate that the global abundance of forest specialists more than halved, on average, from 1970 to 2014. In context, populations of terrestrial species declined globally by an average of 41% over the same time period, which suggests that vertebrates in other terrestrial habitats have fared less badly. However, the population trends among forest specialists remain better than for species living in freshwater habitats, which exhibit more negative population trends [4,55] and a greater risk of extinction [56] than terrestrial counterparts. The result for the forest specialist index was consistent among mammals and herptiles but less so among birds, especially from temperate forests. Differences in average trends between taxonomic groups were significant and, while the effect of threats has not been quantified, the available evidence suggests the negative trend in mammals could be the result of targeted hunting, especially in the tropics [57]. The fungal disease chytridiomycosis, sometimes exacerbated by climate change, could explain

time periods are illustrated by corresponding changes in the index to a slower decline. There is an increase in the percentage of increasing annual trends in 2013 and 2014 and the percentage in 2014 is the highest out of all 44 years; this pattern is notable across all taxa (electronic supplementary material, S7). The average rate of change per species was negative for herptiles and mammals and slightly positive for birds (figure 2), with no overlap between the error bars of each group. This result was echoed when comparing declining and increasing years. There were more declining years than increasing among species trends for mammals (53% of all annual data points) and herptiles (63% of all annual data points); the reverse was true for birds, where there were more increasing years (52% of all annual data points). For all taxa, the percentage of increasing and

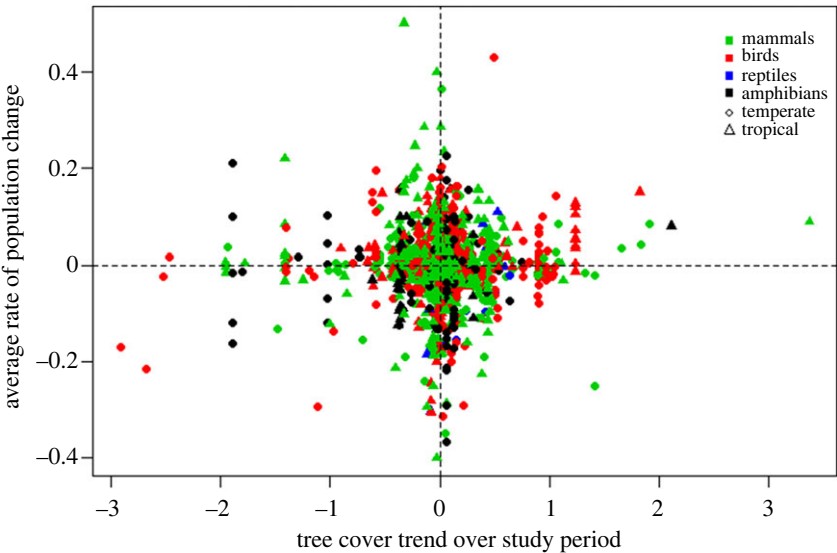

**Figure 3.** Average rate of change of forest vertebrate populations (specialists and generalists) with abundance data covering at least a 5-year range between 1982 and 2016 from the LPD, and tree cover trend within a 5 km radius of each population's study location calculated over the same period as the population data from remotely sensed tree cover data [8]. Green, mammals; red, birds; blue, reptiles; black, amphibians. Circles, temperate biomes; triangles, tropical biomes. *N*=1668. (Online version in colour.)

the stronger negative result for herptiles (e.g. [58,59]). Abundance trends are worse in the tropics, as might be expected, given the more rapid rates of forest loss in tropical regions [21] over that period. The final years of the index, 2013 and 2014, showed an increasing trend as a result of a greater proportion of increasing annual trends among species than in previous years, across all taxa. As there have been other increasing trend years in the index throughout the time series followed by a decline (1991–1992, 2001–2002, 2004–2006), it is not possible to say at this stage whether the latest upturn in the Forest Specialist Index is a sign of a significant, longer-term increase in the abundance of forest specialists.

In understanding the overall reduction in the rate of decline of the index after 2000, we need to consider three factors that are pertinent to interpreting trends in composite indices: species with increasing trends entering the dataset, species with declining trends leaving the dataset and improvement in species trends from declining to increasing or stable during this time period. The first two factors result from turnover in the species data that contributes to the index as data are not available for all 44 years for all species. This turnover in data is observed in our dataset: for example, between 2000 and 2002, data for 12 declining and four increasing species ended at the same time as data for 10 increasing and four declining species entered the dataset. This type of change in the dataset suggests that the reduced rate of decline may not entirely reflect overall improving status for species in the dataset, rather a change in the underlying data coupled with some species recoveries. This highlights a limitation of composite indices such as this where the temporal representation of species data is not comprehensive across the time series [60] and illustrates the need for diagnostics to accompany interpretation as well as additional data to strengthen the index. In addressing the third factor, and in order to eliminate any effect of data turnover, we looked at species with data present in all decades. These are predominantly bird species from the Nearctic, which are well monitored over the long term. After an initial

decline, the average trend for this set of species does show an improvement to stability from the mid-2000s, but this trend is not yet increasing (electronic supplementary material, S13). The stabilization of trends in forest bird species in the Nearctic is consistent with other findings [61]. It is worth noting that species biodiversity data are currently skewed away from where species richness is greatest [62], limiting our ability to identify and address threats in some of the most biodiverse areas on the planet. The lack of population time series in the LPD from forest hotspots in Africa, Asia and the Amazon highlights this issue. To develop a more representative picture of the status of forest biodiversity and drivers of population change, these data gaps need to be filled. This will require greater investment in systematic, long-term, on-the-ground monitoring of forest vertebrates and improved data sharing within the research community.

While remote sensing allows quantitative monitoring of forest cover change, limitations are to be expected in its use for monitoring forest populations: processes of defaunation are more cryptic and difficult to track [2], even occurring in large protected habitats [63]. The use of remote sensing to inform assessments of extinction risk for forest-dependent species has been demonstrated [64]. However, the relationship between habitat change and population change is not necessarily linear and the influence of threats other than habitat loss could also be important, which means that a species-specific approach may need to be taken when using habitat or land cover change to inform the status of a species [64,65]. Our results provide evidence that a satellite-derived assessment of forest cover change alone is inadequate as an indicator of trends in forest biodiversity. We did not find significant evidence of a consistent relationship between forest vertebrate populations and tree cover change in the surrounding area. Further, discrepancies between satellite-derived tree cover datasets in estimates of tree cover change or area indicate the uncertainties associated with tree cover assessments [40,41]. Analyses such as these would benefit from a global, systematically developed dataset categorizing forest areas into natural or planted forests, with temporal

information detailing when each plantation was established. This would allow tree cover losses or gains within plantations to be identified, allowing for more rigorous checks of the relationship between populations of forest-dwelling species and natural forest cover change.

Our finding of exploitation as a key driver of forest specialist population decline supports evidence presented elsewhere. An analysis of threat information for 8688 species on the IUCN Red List of Threatened Species identified overexploitation alongside agriculture (principally crop and livestock farming) as the main drivers of biodiversity loss [42]. The intensification of climate and other global environmental changes is predicted to interact with overexploitation and other pressures to lead to severe future degradation of tropical forests unless alternative, non-destructive development pathways are followed [12]. With most drivers of change interacting in space, time and organizational level [66], an explicitly linked set of forest biodiversity indicators may be more useful than reliance on any individual indicator to understand and communicate forest biodiversity trends and guide policy [67].

The Forest Specialist Index should be among such a set of indicators. This indicator has now been put forward through the Biodiversity Indicators Partnership to measure progress towards Aichi Targets 5, 7 and 12 (https://www.bipindicators.net/indicators/living-planet-index/living-planet-index-forest-specialists) and would complement existing indicators in monitoring progress towards SDG 15, the post-2020 framework under the CBD and in the delivery of the Paris Agreement. As such, it would also be a valuable inclusion in the Global Core Set of forest-related indicators as being coordinated by the FAO.

The findings presented here also demonstrate the importance of complementing satellite-derived datasets with repeated on-the-ground species surveys and site-specific threat information when assessing the status and drivers of forest biodiversity, as advocated for elsewhere [68–70]. While remote sensing data have undoubtedly improved our ability to independently monitor and assess changes in forest cover, there are many additional drivers of forest population change that can only be identified by looking below the canopy. A focus on forest cover change alone risks masking below-canopy processes, such as defaunation, with grave consequences not only for forest biodiversity but also long-term forest health and carbon storage [24,27,28]. Therefore, we must not lose sight of the crucial role that site-level species monitoring plays in understanding trends and drivers of forest biodiversity change.

Data accessibility. The vertebrate population data were taken from the Living Planet Database which is hosted online at www.livingplanetindex.org. The data used for the analysis are available in the electronic supplementary material. Part of the dataset includes confidential data which have been shared under an agreement and are not publicly available. The species details, location and reference have been anonymized and the raw population data replaced with modelled population lambda values. The Forest Specialists Index was calculated using the R package rlpi available at https://github.com/Zoological-Society-of-London/rlpi.

Authors' contributions. E.J.G. and L.M. carried out the statistical analyses with guidance from R.F., M.B.J.H. and S.L.L.H. W.B.-C. and W.D.S. conceived and coordinated the study. All authors contributed to the drafting of the manuscript. All authors gave final approval for publication and agree to be held accountable for the work performed herein.

Competing interests. We declare we have no competing interests.

Funding. The work of E.J.G., M.B.J.H., S.L.L.H. and W.D.S. was funded and supported by WWF-UK, WWF-Germany and WWF-France. Institutional support was provided by WWF-UK to the work of W.B.-C., and ZSL Institute of Zoology provided institutional support to the work of L.M. and R.F.

Acknowledgements. We thank the following collaborators from WWF: Pablo Pacheco, Karen Mo, Lucy Young and Mark Wright for reviewing and discussing the development of this research, as well as Susanne Winter and Daniel Vallauri for supporting the research. We also thank Jack Plummer for conducting a preliminary analysis of the index whilst volunteering at ZSL, and Emma Martin at UNEP-WCMC for help with data collection.

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
