## [Reviewer comments · Proceedings of the Royal Society B: Biological Sciences]

Review History

RSPB-2019-1785.R0 (Original submission)

Review form: Reviewer 1

Recommendation

Reject – article is not of sufficient interest (we will consider a transfer to another journal)

Scientific importance: Is the manuscript an original and important contribution to its field?

Good

General interest: Is the paper of sufficient general interest?

Acceptable

Quality of the paper: Is the overall quality of the paper suitable?

Acceptable

Is the length of the paper justified?

Yes

Should the paper be seen by a specialist statistical reviewer?

No

Do you have any concerns about statistical analyses in this paper? If so, please specify them explicitly in your report.

Yes

It is a condition of publication that authors make their supporting data, code and materials available - either as supplementary material or hosted in an external repository. Please rate, if applicable, the supporting data on the following criteria.

Is it accessible?

Yes

Is it clear?

Yes

Is it adequate?

No

Do you have any ethical concerns with this paper?

No

Comments to the Author

This manuscript describes the production of a Living Planet Index for forest species. It extracts trend data for forest species from the existing LPI database. The manuscript then goes on to compare these population trends with tree cover change assessed from satellite remote sensing data at the relevant location. Although no overall correlation between species population trends and tree cover change trends are found the authors do suggest that there are relationships for a large number (c 40%) of species examined. They then go on to undertake a simple description of the characteristics of the species for which relationships are detected, albeit without comparison back to the characteristics of species for which no relationships were detected.

The production of the LPI for forest species and a comparison between observed trends and satellite remote sensing tree cover data is an original and worthwhile topic for a manuscript. As such the manuscript has much to commend it. However, there are some areas that need to be considered.

Introduction

In the introduction the authors note that various international agreements and conventions use forest cover as a proxy for forest biodiversity. I feel this is a simplification as nowhere does it say that, for example, CBD uses forest cover as a measure of forest biodiversity. The reference in Target 5 to 'other habitats' instead indicates it is being used as a measure of forest habitats and not biodiversity. Perhaps some re wording is needed.

Citation needed to support statement in line 87 about carbon and health and line 90 on meeting the 1.5 degrees climate change target.

On line 109 it would be good to clarify why the authors hypothesise that forest biodiversity is declining. There is information in the introduction about forest loss, but this relates to forest loss not species. I think there is a missing link here to get to the hypothesis.

Methods

Line 191 - are the two measures of tree cover loss correlated? Errors in the tree cover data could explain an absence of correlation with observed species trend. It would be good to know if the tree data agree at least. At the very least add some of the accuracy assessments for the Song data

from their analysis to help the reader assess how accurate such 5 km data are, but for preference add a correlation between change on the site studies here. This is of particular importance given that you then note in lines 214 - 216 that there was considerable fluctuation in tree loss estimates. You need to convince the reader that these 5 km data really reliable and appropriate for this analysis.

Line 195 - the Song et al and Hansen et al data refer to tree cover, not forest cover. The consistence of use of tree cover should be checked through the manuscript and standardised. You make this point in line 206, but i you need to be consistent. Also, please be consistent in the use of negative or inverse correlations in the manuscript.

Line 224 - please give a summary of the methods of Spooner et al so readers who do not have access to it can understand what you did.

In the calculation of these figures of population change did you consider weighting by a transformed SE in the subsequent analysis? This is reasonably common place in these types of analysis and ensures that the better relationships carry more weight into the analysis. The same applies to tree cover change estimates (ie those derived from the beta for the year effect). Also on the statistical analysis why did you consider variables individually rather than in a multi variate model? I can foresee potential interactions between variables that might be good to examine. For example, a decrease in tree cover by 50% might be expected to have a different impact if initial cover was 100% than if initial cover was 50%.

Results

Figure 1 is the key result here, but I feel that the presentation of geographical / taxonomic break downs in the same format as the LPI graph would be more valuable. This would enable a rapid comparison of the trends in, for example, the Nearctic with Africa. Presentation of this trend, and the errors around it, would help in the geographical interpretation of the results. Additionally, could the LPI trend for all species be added for comparison?

The tables summarizing the models raised some questions for me. First is why the authors use AIC and statistical significance. This is odd as these are two philosophies for assessing variable importance. I suggest they pick one or the other and not both. This is particularly important when the results conflict - In table 1 there is apparently a significant relationship between change and body mass ($P = 0.002$), but the associated AIC value indicates that AIC increases, suggesting the addition of the variable did not improve the model. This needs to be explained.

In Table 2 and subsequent tables, how was delta AIC generated for the null model? Was this in fact not the null model but the model with the random effect for location added? In which case please give the original AIC too for the null model. Please give the direction of the change in AIC too to improve the clarity of tables. Finally, the parameter estimates are often small - there is little value in giving a value of 0.00. Could you rescale the variables so that you can quote informative parameter estimates.

Line 341 - I am not sure these are drivers as much as correlates. Please change wording.

On the cross correlation / literature review part of the analysis I feel that more information is needed to give the results more context. First, while I see the logic behind the assessment of positive and negative correlations with cover change (ie the use of $2*\sqrt{n}$), I feel that there are more robust methods that could be used that make better use of the data (see above in comments on methods). I also think that the numerical comparisons presented in this section of the results could be improved upon, even with % bar graphs rather than text describing results. Importantly, how many of the species for which no relationship (positive or negative) were threatened by the pressures listed? You only present information for the 71 with relationships, but we need to know how many with no relationships were also threatened to put the results in context.

Discussion

The discussion is generally well framed, but consideration of some of the points raised above, especially the threats to the species for which no correlation with loss was detected, as noted above, might change some of the content of this discussion.

In presenting the forest LPI I was hoping there would be a comparison with the overall LPI, whether visually, or just in text. Ideally the LPI would be added to figure 1 so readers can make the comparison. Even better would be a statistical comparison of the LPI and forest LPI. This prospect is raised on line 140.

The contention that satellite data alone are insufficient to monitor biodiversity is not new, but is worth stressing as done here to ensure that conservation monitoring is complete. One question that does come to mind here though is what proportion of the world's forests / forest species are covered by the LPI compared to the potential of remote sensing for tracking / highlighting concerns across the globe? Until resources are available for more complete field based monitoring is it not better that we have some metric of the global situation? The need for better monitoring, as mentioned in lines 435 - 436 is to me a key point of the paper and should be repeated in the closing paragraph.

Review form: Reviewer 2

Recommendation

Accept with minor revision (please list in comments)

Scientific importance: Is the manuscript an original and important contribution to its field?

Good

General interest: Is the paper of sufficient general interest?

Good

Quality of the paper: Is the overall quality of the paper suitable?

Good

Is the length of the paper justified?

Yes

Should the paper be seen by a specialist statistical reviewer?

No

Do you have any concerns about statistical analyses in this paper? If so, please specify them explicitly in your report.

No

It is a condition of publication that authors make their supporting data, code and materials available - either as supplementary material or hosted in an external repository. Please rate, if applicable, the supporting data on the following criteria.

Is it accessible?

N/A

Is it clear?

Yes

Is it adequate?

Yes

Do you have any ethical concerns with this paper?

No

Comments to the Author**GENERAL COMMENTS:**

In this paper the authors follow the LPI methodology to develop a global indicator of forest specialist vertebrates to detect population trends over time (1970-2014). They also assess the influence of potential drivers of forest vertebrate population change, and compare trends across species groups, realms and biomes. The results show significant declines in forest vertebrate populations over the study period, but no consistent effects of most of the predictors analysed (except for 'over-exploitation'). The paper is well written and generally clear. The analyses seem robust. The resulting forest specialist index seems useful. The lack of association of population trends with potential drivers of change supports the conclusion that forest area (alone) is a poor proxy of forest biodiversity status, and that more detailed (e.g. forest quality) information is needed. Below I provide some specific comments which I hope the authors will find useful to improve the quality of their paper.

SPECIFIC COMMENTS:

L31 - this line is repetitive with L 25, I suggest removing it.

L59 - fragmentation is a result of habitat loss and can be caused by many drivers. I suggest rewording / being more specific (e.g. fragmentation caused by roads).

L78 - 'the pertinence of forest area as a proxy indicator of forest biodiversity has never been tested'. Presumably the authors mean on a global scale? There have been lots of studies relating forest area to biodiversity (though not on a global scale as far as am aware).

L87 - is a citation missing at the end of this sentence?

L120 - what does 'traceable' mean in this context?

L141 - 'each taxa' ... I think this should be the singular 'taxon'

L147 - change 'tends' to 'tend'

L165 - why were forest generalists included here, but not in the forest specialist index described above? Please clarify/justify.

L184 - what's the temporal scale of the 'shorter-term' dataset?

L195-207 - how was this mismatch in the inclusion of 'plantations' dealt with in analyses and interpreted? I.e forest area datasets include plantations, but the forest specialists index does not (stated in L134)

L209 - some text / references are needed to justify the use of 5 km radii.

L214-218 - this text is a bit unclear and might benefit from some rewording.

L223 - '≥2 years of data covering a ≥5 year period' ... this is unclear, please reword/clarify.

L246-247 - please add supporting references for this or clarify if it's a hypothesis-driven sentence.

L292 - please clarify the rationale behind repeating the analyses with the genus omitted.

L294 - this subtitle seems to refer to text in L314-321 only; I suggest moving further down and renaming the section in L296-312.

L310-311 - a citation is needed for this method and why it's appropriate in this case.

L314-321 - was this a qualitative assessment only?

L317 - I thought plantation forests were not included; please clarify.

L338-340 - in SM 6, in addition to total number shown at the moment, it would be useful to see the percentage of species in each category (increasing, decreasing, stable) for each realm & biome.

L355 - I'm not used to seeing negative AIC values. It would perhaps be useful to clarify (for other readers like myself) that a 'more negative' AIC value indicates a better model fit (if this is indeed the case?)

L358 - although according to table 2 delta AIC for this model was zero (so no model improvement)... it's hard to figure out what criteria the authors are using in each instance (p value, confidence interval, AIC); this should be made clearer and ensure consistency.

L403 – It would be useful to put these results in a wider context by commenting on how trends forest vertebrate populations compare to overall LPI, e.g. are they faring better/worse/similar?

L411 – I suggest rephrasing to ‘...whether the latest upturn in the FSI is a sign of a significant, longer-term increase in the abundance of...’

L416 – and from declining to stable?

L418 – In what way is the data turnover taking place? E.g. x declining species ‘leaving’ the dataset vs x increasing species ‘entering’...

L458-466 – I suggest including a few sentences about the potential mechanisms driving some of these trends.

Decision letter (RSPB-2019-1785.R0)

13-Sep-2019

Dear Dr Hill:

I am writing to inform you that your manuscript RSPB-2019-1785 entitled "Below the canopy: global trends in forest vertebrate populations and their drivers" has, in its current form, been rejected for publication in Proceedings B.

This action has been taken on the advice of referees and the AE, who have recommended that substantial revisions are necessary (please see their detailed comments, below). With this in mind we would be happy to consider a resubmission, provided the comments of the referees are fully addressed. However please note that this is not a provisional acceptance and we will send your revised manuscript back out for review.

- 1) A ‘response to referees’ document including details of how you have responded to the comments, and the adjustments you have made.
- 2) A clean copy of the manuscript and one with 'tracked changes' indicating your 'response to referees' comments document.
- 3) Line numbers in your main document.

Sincerely,
 Dr Sarah Brosnan
 Editor, Proceedings B
<mailto:proceedingsb@royalsociety.org>

Associate Editor

Board Member: 1

Comments to Author:

The study by Hill et al. uses the Living Planet Index methodology to develop an index for forest specialist vertebrates and explore its changes over time. It then examines the underlying drivers of the observed temporal trends with specific focus on forest cover. This paper has been reviewed by two referees. While reviewer 2 was more positive, reviewer 1 had a number of serious concerns about the methodology of the study, which I share. Both reviewers questioned the claim by the authors that forest cover is commonly used as a proxy for forest biodiversity while the relationship between the two has never been tested. This claim needs to be substantiated by references. It is also important to distinguish between relationship between forest cover and biodiversity and changes in forest cover and biodiversity. Both were tested in this study, but there is no reason why both should show the same trends, e.g. if there is a threshold of % forest cover above which changes in forest cover are not significantly affecting biodiversity, then different outcome might be expected (see comment by reviewer 1 about the importance of exploring the interactions between predictors instead of considering predictors individually). Another problem picked up by the reviewer 1 is that for the second part of their analyses, the authors have focused only on 71 populations for which significant relationship with forest cover was found. This resulted in exclusion of 104 populations with no correlation with forest cover from the analysis. The reviewer pointed out that it would be also very important to know the characteristics of species for which no relationship with forest cover has been detected and how many of these species are also threatened by the various pressures examined. In addition, this reviewer has questioned the methodology used for categorizing cross-correlations into positive, negative or uncorrelated with forest cover, which is very much akin to vote-counting. I share this concern. The $2\sqrt{n}$ approach needs to be better justified and ideally a more robust methodology used. Overall, while the study addresses an important topic, is timely and well written, I am afraid due to the identified methodological shortcomings it is not acceptable for publication in Proc Roy Soc B in its current form. If the authors are able to address the comments raised by the referee, the study might become acceptable, but it would require significant changes to the methodology and analyses presented, which is beyond the revision option.

Reviewer(s)' Comments to Author:

Referee: 1

Comments to the Author(s)

This manuscript describes the production of a Living Planet Index for forest species. It extracts trend data for forest species from the existing LPI database. The manuscript then goes on to compare these population trends with tree cover change assessed from satellite remote sensing data at the relevant location. Although no overall correlation between species population trends and tree cover change trends are found the authors do suggest that there are relationships for a large number (c 40%) of species examined. They then go on to undertake a simple description of the characteristics of the species for which relationships are detected, albeit without comparison back to the characteristics of species for which no relationships were detected.

The production of the LPI for forest species and a comparison between observed trends and satellite remote sensing tree cover data is an original and worthwhile topic for a manuscript. As such the manuscript has much to commend it. However, there are some areas that need to be considered.

Introduction

In the introduction the authors note that various international agreements and conventions use forest cover as a proxy for forest biodiversity. I feel this is a simplification as nowhere does it say that, for example, CBD uses forest cover as a measure of forest biodiversity. The reference in

Target 5 to 'other habitats' instead indicates it is being used as a measure of forest habitats and not biodiversity. Perhaps some re wording is needed.

Citation needed to support statement in line 87 about carbon and health and line 90 on meeting the 1.5 degrees climate change target.

On line 109 it would be good to clarify why the authors hypothesise that forest biodiversity is declining. There is information in the introduction about forest loss, but this relates to forest loss not species. I think there is a missing link here to get to the hypothesis.

Methods

Line 191 - are the two measures of tree cover loss correlated? Errors in the tree cover data could explain an absence of correlation with observed species trend. It would be good to know if the tree data agree at least. At the very least add some of the accuracy assessments for the Song data from their analysis to help the reader assess how accurate such 5 km data are, but for preference add a correlation between change on the site studies here. This is of particular importance given that you then note in lines 214 - 216 that there was considerable fluctuation in tree loss estimates. You need to convince the reader that these 5 km data really reliable and appropriate for this analysis.

Line 195 - the Song et al and Hansen et al data refer to tree cover, not forest cover. The consistence of use of tree cover should be checked through the manuscript and standardised. You make this point in line 206, but i you need to be consistent. Also, please be consistent in the use of negative or inverse correlations in the manuscript.

Line 224 - please give a summary of the methods of Spooner et al so readers who do not have access to it can understand what you did.

In the calculation of these figures of population change did you consider weighting by a transformed SE in the subsequent analysis? This is reasonably common place in these types of analysis and ensures that the better relationships carry more weight into the analysis. The same applies to tree cover change estimates (ie those derived from the beta for the year effect). Also on the statistical analysis why did you consider variables individually rather than in a multi variate model? I can foresee potential interactions between variables that might be good to examine. For example, a decrease in tree cover by 50% might be expected to have a different impact if initial cover was 100% than if initial cover was 50%.

Results

Figure 1 is the key result here, but I feel that the presentation of geographical / taxonomic break downs in the same format as the LPI graph would be more valuable. This would enable a rapid comparison of the trends in, for example, the Nearctic with Africa. Presentation of this trend, and the errors around it, would help in the geographical interpretation of the results. Additionally, could the LPI trend for all species be added for comparison?

The tables summarizing the models raised some questions for me. First is why the authors use AIC and statistical significance. This is odd as these are two philosophies for assessing variable importance. I suggest they pick one or the other and not both. This is particularly important when the results conflict - In table 1 there is apparently a significant relationship between change and body mass ($P = 0.002$), but the associated AIC value indicates that AIC increases, suggesting the addition of the variable did not improve the model. This needs to be explained.

In Table 2 and subsequent tables, how was delta AIC generated for the null model? Was this in fact not the null model but the model with the random effect for location added? In which case

please give the original AIC too for the null model. Please give the direction of the change in AIC too to improve the clarity of tables. Finally, the parameter estimates are often small – there is little value in giving a value of 0.00. Could you rescale the variables so that you can quote informative parameter estimates.

Line 341 – I am not sure these are drivers as much as correlates. Please change wording.

On the cross correlation / literature review part of the analysis I feel that more information is needed to give the results more context. First, while I see the logic behind the assessment of positive and negative correlations with cover change (ie the use of $2*\sqrt{n}$), I feel that there are more robust methods that could be used that make better use of the data (see above in comments on methods). I also think that the numerical comparisons presented in this section of the results could be improved upon, even with % bar graphs rather than text describing results. Importantly, how many of the species for which no relationship (positive or negative) were threatened by the pressures listed? You only present information for the 71 with relationships, but we need to know how many with no relationships were also threatened to put the results in context.

Discussion

The discussion is generally well framed, but consideration of some of the points raised above, especially the threats to the species for which no correlation with loss was detected, as noted above, might change some of the content of this discussion.

In presenting the forest LPI I was hoping there would be a comparison with the overall LPI, whether visually, or just in text. Ideally the LPI would be added to figure 1 so readers can make the comparison. Even better would be a statistical comparison of the LPI and forest LPI. This prospect is raised on line 140.

The contention that satellite data alone are insufficient to monitor biodiversity is not new, but is worth stressing as done here to ensure that conservation monitoring is complete. One question that does come to mind here though is what proportion of the world's forests / forest species are covered by the LPI compared to the potential of remote sensing for tracking / highlighting concerns across the globe? Until resources are available for more complete field based monitoring is it not better that we have some metric of the global situation? The need for better monitoring, as mentioned in lines 435 – 436 is to me a key point of the paper and should be repeated in the closing paragraph.

Referee: 2

Comments to the Author(s)

GENERAL COMMENTS:

In this paper the authors follow the LPI methodology to develop a global indicator of forest specialist vertebrates to detect population trends over time (1970-2014). They also assess the influence of potential drivers of forest vertebrate population change, and compare trends across species groups, realms and biomes. The results show significant declines in forest vertebrate populations over the study period, but no consistent effects of most of the predictors analysed (except for 'over-exploitation'). The paper is well written and generally clear. The analyses seem robust. The resulting forest specialist index seems useful. The lack of association of population trends with potential drivers of change supports the conclusion that forest area (alone) is a poor proxy of forest biodiversity status, and that more detailed (e.g. forest quality) information is needed. Below I provide some specific comments which I hope the authors will find useful to improve the quality of their paper.

SPECIFIC COMMENTS:

L31 – this line is repetitive with L 25, I suggest removing it.

L59 – fragmentation is a result of habitat loss and can be caused by many drivers. I suggest rewording / being more specific (e.g. fragmentation caused by roads).

L78 – ‘the pertinence of forest area as a proxy indicator of forest biodiversity has never been tested’. Presumably the authors mean on a global scale? There have been lots of studies relating forest area to biodiversity (though not on a global scale as far as am aware).

L87 – is a citation missing at the end of this sentence?

L120 – what does ‘traceable’ mean in this context?

L141 – ‘each taxa’ ... I think this should be the singular ‘taxon’

L147 – change ‘tends’ to tend’

L165 – why were forest generalists included here, but not in the forest specialist index described above? Please clarify/justify.

L184 – what’s the temporal scale of the ‘shorter-term’ dataset?

L195-207 – how was this mismatch in the inclusion of ‘plantations’ dealt with in analyses and interpreted? I.e forest area datasets include plantations, but the forest specialists index does not (stated in L134)

L209 – some text / references are needed to justify the use of 5 km radii.

L214-218 – this text is a bit unclear and might benefit from some rewording.

L223 – ‘≥2 years of data covering a ≥5 year period’ ... this is unclear, please reword/clarify.

L246-247 – please add supporting references for this or clarify if it’s a hypothesis-driven sentence.

L292 – please clarify the rationale behind repeating the analyses with the genus omitted.

L294 – this subtitle seems to refer to text in L314-321 only; I suggest moving further down and renaming the section in L296-312.

L310-311 – a citation is needed for this method and why it’s appropriate in this case.

L314-321 – was this a qualitative assessment only?

L317 – I thought plantation forests were not included; please clarify.

L338-340 – in SM 6, in addition to total number shown at the moment, it would be useful to see the percentage of species in each category (increasing, decreasing, stable) for each realm & biome.

L355 – I’m not used to seeing negative AIC values. It would perhaps be useful to clarify (for other readers like myself) that a ‘more negative’ AIC value indicates a better model fit (if this is indeed the case?)

L358 – although according to table 2 delta AIC for this model was zero (so no model improvement)... it’s hard to figure out what criteria the authors are using in each instance (p value, confidence interval, AIC); this should be made clearer and ensure consistency.

L403 – It would be useful to put these results in a wider context by commenting on how trends forest vertebrate populations compare to overall LPI, e.g. are they faring better/worse/similar?

L411 – I suggest rephrasing to ‘...whether the latest upturn in the FSI is a sign of a significant, longer-term increase in the abundance of...’

L416 – and from declining to stable?

L418 – In what way is the data turnover taking place? E.g. x declining species ‘leaving’ the dataset vs x increasing species ‘entering’...

L458-466 – I suggest including a few sentences about the potential mechanisms driving some of these trends.

Author's Response to Decision Letter for (RSPB-2019-1785.R0)

See Appendix A.

RSPB-2020-0533.R0

Review form: Reviewer 1

Recommendation

Accept with minor revision (please list in comments)

Scientific importance: Is the manuscript an original and important contribution to its field?

Excellent

General interest: Is the paper of sufficient general interest?

Excellent

Quality of the paper: Is the overall quality of the paper suitable?

Excellent

Is the length of the paper justified?

Yes

Should the paper be seen by a specialist statistical reviewer?

No

Do you have any concerns about statistical analyses in this paper? If so, please specify them explicitly in your report.

No

It is a condition of publication that authors make their supporting data, code and materials available - either as supplementary material or hosted in an external repository. Please rate, if applicable, the supporting data on the following criteria.

Is it accessible?

Yes

Is it clear?

Yes

Is it adequate?

Yes

Do you have any ethical concerns with this paper?

No

Comments to the Author

This is an interesting and timely manuscript that attempts to highlight the need for field collected data in monitoring. As we head into the next CBD cycle, we need informative studies such as this to help identify relevant indicators that link to biodiversity. This manuscript describes an indicator that could be complimentary to the satellite image derived assessments of forest change. The manuscript does need some areas of clarification in addition to the specific comments below:

The authors need to be clear whether they are looking at forest cover from remote sensing data or forest change. The compare forest change with index change, but in the text refer to both forest cover and cover change. Consequently, it is unclear to me what they want the reader to focus on. The matter is confused more by their presenting a correlation between forest cover from Song et al and Hanse for 2010, but not forest change. I see this was requested in the previous set of

reviews too (1.4) but was not implemented. It is essential that they present a correlation of forest change between a comparable time period (2000 – 2010) for their study sites using both sets of satellite image data. Without this it is not possible to assess how consistent the measures of change are. Related to this, I disagree with sentence lines 103 to 105 which is a fundamental point of the paper language but not content – you look at change in indices. Forest cover will not correlate with FSI for obvious reasons (100 ha boreal forest will have very different FSI than 100 ha humid tropical forest), but instead it might correlate with change.

This brings me to another point which is would we expect change in 10 ha of one to result in change of similar magnitude in biodiversity in another. Fitting interaction terms to the models of FSI and satellite data would help examine this effect.

The authors need to present a better explanation of why they chose the correlates that they did (line 262) as currently none given. Otherwise it appears an eclectic pick and mix selection. Was the selection based on previous studies or assessment of level 2 or 3 IUCN threats?

Finally, in their summation, the authors correctly and importantly state the need to have field measurements of biodiversity rather than relying on remote sensing alone. They are not the first to state this, and perhaps some recognition that this is oft stated would balance the conclusions a bit.

Specific comments.

Title – It's a personal thing, but I do not like titles with colons as one part of the title can almost always be deleted. I suggest deleting below the canopy here.

Ln 20 – you look at the correlation with forest change first – change wording to note this.

Ln22 – It is not effects of satellite derived tree cover trends, it is a correlation between SPI and tree cover from sat data that you look at. There is a very important difference. Ln 25 is okay.

Ln 27 Monitoring alone will not allow forest species to recover. Much more must be done.

Change wording here to reflect fact that monitoring allows us to track recovery and implement adaptive management or wording along those lines

Ln35 – Not really a deadline, rather review date.

Ln 35 – delete legally established unless in FAO wording – it would be interesting to see an illegally established protected area

Ln 86 – use °C

Ln 90 – delete 'Recent' as it is a time limited term in this case

Ln 102 – derived from rather than using as you use correlation to assess the relationship

Ln 106 – 109. Point 1 is not a hypothesis as currently worded.

Ln 119 – Data are, not data is. Use if 'is' jars when reading. Also Ln 179.

Ln 180 – six, not 6

Ln 184 – what is a biologically useful resolution? Is it a resolution that is appropriate to the size of your study sites from which the population data came? That is probably a more important reason.

Ln 218 – 222 – this section appears to be out of place and should be in the additional correlates section.

LN 253 – 255 – this is a massive reduction in the number of populations used. You drop specialists by 200 for Hansen based on my calculation. Is there a reason for this as to me this drop indicates a serious issue with these data? I think you need to add more of an explanation as to why you go from 268 (Ln 137) to 74. It might be I have missed something in the text.

Ln 257 – 260 – This is a measure of correlation in area, but you are looking at change in the analysis. Would it be possible to look at the correlation between change 2000 – 2010 in both datasets as a more relevant measure of agreement?

Ln 311 – delete 'wished' and reword, but otherwise good to see the inclusion of this analysis here.

Ln 342 – Are these SE or 95% CIs around parameter estimates? The non-overlapping nature of the bars in Figure 2 is interesting and worth mentioning. It isn't mentioned anywhere in the manuscript, but is the massive decline in herptiles due to Chytrid?

Figure 3 – given the notable differences in trends in figure 2 b and c, I think it is useful to use different symbols for either realm or biome in this figure (as long as it is not a mess) as I am curious to see what this looks like.

Ln 387 – on average halved or just halved?

Ln 431- The use of satellite remote sensing data for monitoring populations is often caveated with this very point. While there is no need to undertake a thorough review of the literature it might be worth stressing this point.

Ln 436 – So what is the implication of this for updating red list categories using remote sensing data (e.g. Tracewski et al 2016)?

Decision letter (RSPB-2020-0533.R0)

30-Mar-2020

Dear Dr Hill:

Your manuscript has now been peer reviewed and the reviews have been assessed by an Associate Editor. The reviewers' comments (not including confidential comments to the Editor) and the comments from the Associate Editor are included at the end of this email for your reference. As you will see, the reviewers and the Editors have raised a few concerns with your manuscript and we would like to invite you to revise your manuscript to address them.

Research ethics:

Use of animals and field studies:

Please submit a copy of your revised paper within three weeks. If we do not hear from you within this time your manuscript will be rejected. If you are unable to meet this deadline please let us know as soon as possible, as we may be able to grant a short extension.

Best wishes,
Dr Sarah Brosnan
Editor, Proceedings B
mailto: proceedingsb@royalsociety.org

Associate Editor Board Member
Comments to Author:

The revised ms has been re-reviewed by one of the original referees. They found the paper improved, but there are still a few points to address. Most of them are minor, but the most important point is to clarify whether the authors are looking at forest cover from remote sensing data or forest change. The reviewer notes that "it is essential that the authors present a correlation of forest change between a comparable time period (2000 – 2010) for their study sites using both sets of satellite image data. Without this it is not possible to assess how consistent the measures of change are". I therefore recommend the revision to address the above point and other comments by the reviewer.

Reviewer(s)' Comments to Author:

Referee: 1

Comments to the Author(s).

This is an interesting and timely manuscript that attempts to highlight the need for field collected data in monitoring. As we head into the next CBD cycle, we need informative studies such as this to help identify relevant indicators that link to biodiversity. This manuscript describes an indicator that could be complimentary to the satellite image derived assessments of forest change. The manuscript does need some areas of clarification in addition to the specific comments below:

The authors need to be clear whether they are looking at forest cover from remote sensing data or forest change. The compare forest change with index change, but in the text refer to both forest cover and cover change. Consequently, it is unclear to me what they want the reader to focus on. The matter is confused more by their presenting a correlation between forest cover from Song et al and Hanse for 2010, but not forest change. I see this was requested in the previous set of reviews too (1.4) but was not implemented. It is essential that they present a correlation of forest change between a comparable time period (2000 – 2010) for their study sites using both sets of satellite image data. Without this it is not possible to assess how consistent the measures of change are. Related to this, I disagree with sentence lines 103 to 105 which is a fundamental point of the paper language but not content – you look at change in indices. Forest cover will not correlate with FSI for obvious reasons (100 ha boreal forest will have very different FSI than 100 ha humid tropical forest), but instead it might correlate with change.

This brings me to another point which is would we expect change in 10 ha of one to result in change of similar magnitude in biodiversity in another. Fitting interaction terms to the models of FSI and satellite data would help examine this effect.

The authors need to present a better explanation of why they chose the correlates that they did (line 262) as currently none given. Otherwise it appears an eclectic pick and mix selection. Was the selection based on previous studies or assessment of level 2 or 3 IUCN threats?

Finally, in their summation, the authors correctly and importantly state the need to have field measurements of biodiversity rather than relying on remote sensing alone. They are not the first to state this, and perhaps some recognition that this is oft stated would balance the conclusions a bit.

Specific comments.

Title – It's a personal thing, but I do not like titles with colons as one part of the title can almost always be deleted. I suggest deleting below the canopy here.

Ln 20 – you look at the correlation with forest change first – change wording to note this.

Ln22 – It is not effects of satellite derived tree cover trends, it is a correlation between SPI and tree cover from sat data that you look at. There is a very important difference. Ln 25 is okay.

Ln 27 Monitoring alone will not allow forest species to recover. Much more must be done.

Change wording here to reflect fact that monitoring allows us to track recovery and implement adaptive management or wording along those lines

Ln35 – Not really a deadline, rather review date.

Ln 35 – delete legally established unless in FAO wording – it would be interesting to see an illegally established protected area

Ln 86 – use °C

Ln 90 – delete 'Recent' as it is a time limited term in this case

Ln 102 – derived from rather than using as you use correlation to assess the relationship

Ln 106 – 109. Point 1 is not a hypothesis as currently worded.

Ln 119 – Data are, not data is. Use if 'is' jars when reading. Also Ln 179.

Ln 180 – six, not 6

Ln 184 – what is a biologically useful resolution? Is it a resolution that is appropriate to the size of your study sites from which the population data came? That is probably a more important reason.

Ln 218 – 222 – this section appears to be out of place and should be in the additional correlates section.

LN 253 – 255 – this is a massive reduction in the number of populations used. You drop specialists by 200 for Hansen based on my calculation. Is there a reason for this as to me this drop indicates a serious issue with these data? I think you need to add more of an explanation as to why you go from 268 (Ln 137) to 74. It might be I have missed something in the text.

Ln 257 – 260 – This is a measure of correlation in area, but you are looking at change in the analysis. Would it be possible to look at the correlation between change 2000 – 2010 in both datasets as a more relevant measure of agreement?

Ln 311 – delete 'wished' and reword, but otherwise good to see the inclusion of this analysis here.

Ln 342 – Are these SE or 95% Cis around parameter estimates? The non-overlapping nature of the bars in Figure 2 is interesting and worth mentioning. It isn't mentioned anywhere in the manuscript, but is the massive decline in herptiles due to Chytrid?

Figure 3 – given the notable differences in trends in figure 2 b and c, I think it is useful to use different symbols for either realm or biome in this figure (as long as it is not a mess) as I am curious to see what this looks like.

Ln 387 – on average halved or just halved?

Ln 431- The use of satellite remote sensing data for monitoring populations is often caveated with this very point. While there is no need to undertake a thorough review of the literature it might be worth stressing this point.

Ln 436 – So what is the implication of this for updating red list categories using remote sensing data (e.g. Tracewski et al 2016)?

Author's Response to Decision Letter for (RSPB-2020-0533.R0)

See Appendix B.

RSPB-2020-0533.R1 (Revision)

Review form: Reviewer 1

Recommendation

Accept as is

Scientific importance: Is the manuscript an original and important contribution to its field?

Excellent

General interest: Is the paper of sufficient general interest?

Good

Quality of the paper: Is the overall quality of the paper suitable?

Excellent

Is the length of the paper justified?

Yes

Should the paper be seen by a specialist statistical reviewer?

No

Do you have any concerns about statistical analyses in this paper? If so, please specify them explicitly in your report.

No

It is a condition of publication that authors make their supporting data, code and materials available - either as supplementary material or hosted in an external repository. Please rate, if applicable, the supporting data on the following criteria.

Is it accessible?

Yes

Is it clear?

Yes

Is it adequate?

Yes

Do you have any ethical concerns with this paper?

No

Comments to the Author

Thank you for giving considerations to my comments and providing clear feedback on how they were addressed.

Decision letter (RSPB-2020-0533.R1)

01-May-2020

Dear Dr Hill

I am pleased to inform you that your manuscript entitled "Below the canopy: global trends in forest vertebrate populations and their drivers" has been accepted for publication in Proceedings B.

Open Access

Paper charges

Sincerely,

Dr Sarah Brosnan

Appendix A

No.	Reviewer	Comment	Response
1.1	1	In the introduction the authors note that various international agreements and conventions use forest cover as a proxy for forest biodiversity. I feel this is a simplification as nowhere does it say that, for example, CBD uses forest cover as a measure of forest biodiversity. The reference in Target 5 to 'other habitats' instead indicates it is being used as a measure of forest habitats and not biodiversity. Perhaps some re wording is needed.	This is a valid comment about the exactitude of the language. As suggested, some rewording has been made, and we now state: 'Due to the importance of forests as habitat for many species, forest area is often employed as an indicator in global agreements and processes aimed at slowing and reversing the decline of biodiversity.' (lines 64-66). We also used "biodiversity relevant indicator" rather than "proxy indicator of forest biodiversity" in lines 74-75. The term "proxy indicator" has also been removed from the abstract
1.2	1	Citation needed to support statement in line 87 about carbon and health and line 90 on meeting the 1.5 degrees climate change target.	Two references have been added to support the statement on biodiversity and forest health: Lewis, S. et al. 2015 and Trumbore, S. et al. (2015) (lines 83-84). A further two references (Harper et al., 2018 and Rojelij et al., 2018) were added to support the second statement (line 87), which has been qualified accordingly
1.3	1	On line 109 it would be good to clarify why the authors hypothesise that forest biodiversity is declining. There is information in the introduction about forest loss, but this relates to forest loss not species. I think there is a missing link here to get to the hypothesis.	Clarification and supporting references added: "(given the decline in area of natural forest over time [Keenan et al. 2015] and the link between habitat loss and biodiversity loss [Brooks et al. 2002])" (lines 106-108)
1.4	1	Line 191 - are the two measures of tree cover loss correlated? Errors in the tree cover data could explain an absence of correlation with observed species trend. It would be good to know if the tree data agree at least. At the very least add some of the accuracy assessments for the Song data from their analysis to help the reader assess how accurate such 5 km data are, but for preference add	Added in a correlation analysis between Hansen and MEASURES tree cover data: "In order to examine the agreement between the two tree cover datasets, we assessed the correlation between the 2010 tree cover estimates as determined from the Hansen dataset and the

		a correlation between change on the site studies here. This is of particular importance given that you then note in lines 214 - 216 that there was considerable fluctuation in tree loss estimates. You need to convince the reader that these 5 km data really reliable and appropriate for this analysis.	Song dataset, for the 685 populations included in the Hansen analyses. The correlation between the two datasets was very high (Pearson correlation coefficient = 0.950; $p < 0.001$).” (lines 257-260)
1.5	1	Line 195 - the Song et al and Hansen et al data refer to tree cover, not forest cover. The consistence of use of tree cover should be checked through the manuscript and standardised. You make this point in line 206, but i you need to be consistent.	Edited to more accurately reflect the use of “forest” terminology: “It is important to note that, while the 30-m dataset used in these analyses comes from the Global Forest Change dataset, neither this nor the Song dataset differentiate between natural, semi-natural or non-natural forests (such as plantations).” (lines 203-205)
1.6	1	Also, please be consistent in the use of negative or inverse correlations in the manuscript.	Correlation analyses have been removed (See comments 1.15/1.16)
1.7	1	Line 224 - please give a summary of the methods of Spooner et al so readers who do not have access to it can understand what you did.	We have amended the text to explain how we obtained these values, referencing an earlier section of the methods as well as the Spooner et al paper: ‘Using the annual logged values from the GAM and linear regression performed earlier, we calculated an average rate of change value per each remaining population as our response variable following Spooner et al. (2018).’ (Lines 241-243)
1.8	1	In the calculation of these figures of population change did you consider weighting by a transformed SE in the subsequent analysis? This is reasonably common place in these types of analysis and ensures that the better relationships carry more weight into the analysis. The same applies to tree cover change estimates (ie those derived from the beta for the year effect).	We didn’t consider weighting by SE for population change, as it would tend to place more weight on consistent trends (increase, decline or stable) and downweight a response to a sudden event that occurs mid-time series e.g. a deforestation event might cause a stable population to decline suddenly or vice versa with a conservation intervention. This

			seemed counterintuitive to what we were testing. We tested the effect of SE as a weight in the forest cover analyses. We took the log-10 transformed inverse of SE of the tree cover trend and fitted this as a weight in the mixed effects models with average rate of change of the population as the dependent variable, tree cover trend as the explanatory variable and a random effect of location. We compared the AIC of these models (generalists + specialists AIC = -3956.419; specialists only AIC = -465.0816) with the AIC of the null model which had the same weighting (generalists + specialists AIC = -3966.277; specialists only AIC = -472.8339) and found no change in the results, i.e. there was no significant difference from the null model.
1.9	1	Also on the statistical analysis why did you consider variables individually rather than in a multivariate model? I can foresee potential interactions between variables that might be good to examine. For example, a decrease in tree cover by 50% might be expected to have a different impact if initial cover was 100% than if initial cover was 50%.	Preliminary analyses on a reduced dataset included multivariate models with interactions (including mean tree cover and tree cover trend), but the results did not support the inclusion of these interactions. We therefore selected to run univariate models on the complete dataset in the first instance, and to build complexity if significant effects were found. After noting we carried out univariate models in the first instance, we have added: "with the aim of fitting multivariate models where evidence of a relationship was found for more than one predictor variable." (lines 296-298)
1.10	1	Figure 1 is the key result here, but I feel that the presentation of geographical / taxonomic break downs in the same format as the LPI graph would be more valuable. This would enable a rapid comparison of	The bar graphs showing positive and negative years have been moved to supplementary materials (SM 10). We tried the reviewer's suggestion of examining

		the trends in, for example, the Nearctic with Africa. Presentation of this trend, and the errors around it, would help in the geographical interpretation of the results	disaggregations of the overall index. However, the variable data availability over time for some of the subsets meant that a robust index could not be produced, making comparison difficult. Instead we took the mean and standard error of the mean of species trends for each of the subsets which we thought made a useful comparison. These have been added as a new figure (Figure 2) into the main body of the manuscript. Text was added to the methods to describe the process for producing this figure (lines 163-164)
1.11	1	Additionally, could the LPI trend for all species be added for comparison?	Yes, this is a good suggestion as we allude to already in the methods. We have added the global terrestrial LPI in Figure 1 as we thought this was the most suitable comparison. The global LPI contains freshwater and marine species so the comparison is perhaps not as appropriate. The text was also amended in line 143 to state that the use of the same method enables the forest index to be compared to the global terrestrial LPI rather than the global LPI. Also we added text in lines 332-335 in the results to describe the terrestrial LPI.
1.12	1	The tables summarizing the models raised some questions for me. First is why the authors use AIC and statistical significance. This is odd as these are two philosophies for assessing variable importance. I suggest they pick one or the other and not both. This is particularly important when the results conflict – In table 1 there is apparently a significant relationship between change and body mass ($P = 0.002$), but the associated AIC value indicates that AIC increases, suggesting the addition of the variable did not improve the model. This needs to be explained.	P-values and AIC were both included because there isn't universal agreement about which is best. However, we agree this can be confusing when results conflict. Therefore we have opted to report only on AIC and all references to P-values in tables and text have been removed.

1.13	1	In Table 2 and subsequent tables, how was delta AIC generated for the null model? Was this in fact not the null model but the model with the random effect for location added? In which case please give the original AIC too for the null model. Please give the direction of the change in AIC too to improve the clarity of tables. Finally, the parameter estimates are often small – there is little value in giving a value of 0.00. Could you rescale the variables so that you can quote informative parameter estimates.	Re. Tables 1 and 2: Previously delta AIC was based on the AIC of the model with the lowest AIC, not (necessarily) the null model. This has been changed so that delta AIC is now based on the difference in AIC between each model and the null model. Direction of change in AIC has been added to delta AIC columns. Regarding the null model, in this case we are interested in the importance of the fixed effects. Therefore the most appropriate null model is the model with all fixed effects removed and only the random effects remaining, to allow for random slopes of, in this case, location. See: Luke, Douglas A. Multilevel modeling. Vol. 143. SAGE Publications, Incorporated, 2019. The parameter estimates have now been presented in scientific notation to make the values more meaningful.
1.14	1	Line 341 – I am not sure these are drivers as much as correlates. Please change wording.	Amended to “correlates” (line 167)
1.15	1	On the cross correlation / literature review part of the analysis I feel that more information is needed to give the results more context. First, while I see the logic behind the assessment of positive and negative correlations with cover change (ie the use of $2\sqrt{n}$), I feel that there are more robust methods that could be used that make better use of the data (see above in comments on methods).	Cross correlation analyses have been removed. This is because we do not have qualitative information extracted for the forest specialist populations that showed no correlation with tree cover and therefore could not address comment 1.16. The correlations were carried out in order to identify the forest populations that were most of interest for the literature review. As the literature review has been removed, there is no need to include the correlation analyses in the paper.
1.16	1	I also think that the numerical comparisons presented in this	As we do not have qualitative information for the species with

		section of the results could be improved upon, even with % bar graphs rather than text describing results. Importantly, how many of the species for which no relationship (positive or negative) were threatened by the pressures listed? You only present information for the 71 with relationships, but we need to know how many with no relationships were also threatened to put the results in context.	no correlation with tree cover, all reference to the lit review and correlation analyses have been removed from the paper.
1.17	1	The discussion is generally well framed, but consideration of some of the points raised above, especially the threats to the species for which no correlation with loss was detected, as noted above, might change some of the content of this discussion.	See comments above.
1.18	1	In presenting the forest LPI I was hoping there would be a comparison with the overall LPI, whether visually, or just in text. Ideally the LPI would be added to figure 1 so readers can make the comparison. Even better would be a statistical comparison of the LPI and forest LPI. This prospect is raised on line 140.	Yes, this is a good suggestion as we allude to already in the methods. We have added the global terrestrial LPI in Figure 1 as we thought this was the most suitable comparison. The global LPI contains freshwater and marine species so the comparison is perhaps not as appropriate. The text was also amended in line 143 to state that the use of the same method enables the forest index to be compared to the global terrestrial LPI rather than the global LPI. Also we added text in lines 332-335 in the results to describe the terrestrial LPI. We didn't feel that a statistical comparison was appropriate given that the forest specialist species are a subset of the terrestrial LPI
1.19	1	The contention that satellite data alone are insufficient to monitor biodiversity is not new, but is worth stressing as done here to ensure that conservation monitoring is complete. One question that does come to mind here though is what proportion of the world's forests / forest species are covered by the	We note there is great value in remote-sensing derived datasets (e.g. "While remote sensing data has undoubtedly improved our ability to independently monitor and assess changes in forest cover..") and we state that these datasets should be

		LPI compared to the potential of remote sensing for tracking / highlighting concerns across the globe? Until resources are available for more complete field based monitoring is it not better that we have some metric of the global situation? The need for better monitoring, as mentioned in lines 435 – 436 is to me a key point of the paper and should be repeated in the closing paragraph.	complemented by on the ground species-level monitoring. Therefore we are not saying they should not be used. Rather we argue that relying on them alone “risks masking below-canopy processes, such as defaunation, with grave consequences not only for forest biodiversity but also long-term forest health and carbon storage”, which supports our point about the need for better monitoring (which we repeat in the closing sentence of the discussion: “Therefore, we must not lose sight of the crucial role that site-level species monitoring plays in understanding trends and drivers of forest biodiversity change.”)
2.1	2	L31 – this line is repetitive with L 25, I suggest removing it.	L31 has been deleted
2.2	2	L59 – fragmentation is a result of habitat loss and can be caused by many drivers. I suggest rewording / being more specific (e.g. fragmentation caused by roads).	Agreed on this point, and the sentence has been changed to “Yet deforestation of tropical forests, reducing their land coverage from 12% to less than 5% (Brandon, 2014), along with their degradation and fragmentation, have resulted from large-scale industrial and local subsistence agriculture (Hosonuma et al., 2012) as well as logging, fires and road building (Lewis, Edwards, & Galbraith, 2016).” (lines 51-55)
2.3	2	L78 – ‘the pertinence of forest area as a proxy indicator of forest biodiversity has never been tested’. Presumably the authors mean on a global scale? There have been lots of studies relating forest area to biodiversity (though not on a global scale as far as am aware).	added “at a global scale” (line 75) and we have also changed “proxy indicator of forest biodiversity” to “biodiversity relevant indicator”, in response to comment 1.1
2.4	2	L87 – is a citation missing at the end of this sentence?	Amended to include citations “(Bello et al., 2015; Osuri et al., 2016)” (line 84)

2.5	2	L120 – what does ‘traceable’ mean in this context?	amended to “and be traceable to the original data source” (line 119)
2.6	2	L141 – ‘each taxa’... I think this should be the singular ‘taxon’	amended to “taxon” (line 144)
2.7	2	L147 – change ‘tends’ to tend’	amended to “tend” (line 150)
2.8	2	L165 – why were forest generalists included here, but not in the forest specialist index described above? Please clarify/justify.	The rationale behind limiting the Forest Specialist Index to only specialist species is provided in lines (125-129). Re. analyses including generalists, we have added: “While the Forest Specialist Index reflects population changes in forest specialists to more accurately reflect ecosystem health, changes in tree cover may also affect populations of forest generalists.” (L171-173)
2.9	2	L184 – what’s the temporal scale of the ‘shorter-term’ dataset?	amended to state the shorter term dataset is for 2000-2017 (line 192)
2.10	2	L195-207 – how was this mismatch in the inclusion of ‘plantations’ dealt with in analyses and interpreted? I.e forest area datasets include plantations, but the forest specialists index does not (stated in L134)	This is an important point and we have clarified the text in L134-136 accordingly. Although we used the strict ‘Forest’ habitat definition, it’s possible that the population occurred all or in part in a plantation. The Red List definition used is the best available that we know of for a species but we have added this important caveat that often only the major habitats are recorded.
2.11	2	L209 – some text / references are needed to justify the use of 5 km radii.	Agreed a justification should be included. Added in: “Range size estimates were not known for every species, but body mass estimates were available (see Additional drivers of forest population change below). We calculated the correlation between body mass and range size (both log-10 transformed) for the species with both estimates available, and found a strong relationship (Pearson correlation coefficient

			= 0.87, $p < 0.001$). We therefore used body mass to predict range size for all populations using the 'predict.lm' function in the 'stats' package (R Core Team, 2018) and calculated the mean range size across all populations as 58.5 km ² , equivalent to a circle with a radius of 4.32 km. We rounded this up and fitted buffers with a 5-km radius around the central coordinates of each forest population." (lines 218-225)
2.12	2	L214-218 – this text is a bit unclear and might benefit from some rewording.	amended for clarity to: "By plotting annual tree cover values from the Song dataset against year, we were able to visually assess the change in tree cover ..." (lines 231-232)
2.13	2	L223 – '≥2 years of data covering a ≥5 year period'... this is unclear, please reword/clarify.	Amended to "In both cases we removed populations that no longer had ≥2 years of data spread over at least a 5 year period" (lines 240-241)
2.14	2	L246-247 – please add supporting references for this or clarify if it's a hypothesis-driven sentence.	Amended to include citation: "(Maxwell, Fuller, Brooks, & Watson, 2016)." (line 272)
2.15	2	L292 – please clarify the rationale behind repeating the analyses with the genus omitted.	Amended to clarify: "We then examined the influential genus to determine the cause and, if the genus was known to be responding to a driver other than those included in our analyses (e.g. disease, poisoning), we repeated our analyses with the genus omitted." (lines 318-320)
2.16	2	L294 – this subtitle seems to refer to text in L314-321 only; I suggest moving further down and renaming the section in L296-312.	The lit review and cross-correlation analyses have been removed.
2.17	2	L310-311 – a citation is needed for this method and why it's appropriate in this case.	The lit review and cross-correlation analyses have been removed.
2.18	2	L314-321 – was this a qualitative assessment only?	The lit review and cross-correlation analyses have been removed.
2.19	2	L317 – I thought plantation forests were not included; please clarify.	The lit review and cross-correlation analyses have been removed.
2.20	2	L338-340 – in SM 6, in addition to total number shown at the moment,	The table in SM6 has been expanded to include the

		it would be useful to see the percentage of species in each category (increasing, decreasing, stable) for each realm & biome.	numbers of species in each trend category in each realm and biome
2.21	2	L355 – I'm not used to seeing negative AIC values. It would perhaps be useful to clarify (for other readers like myself) that a 'more negative' AIC value indicates a better model fit (if this is indeed the case?)	amended to include "a more negative AIC indicates a better model fit;" (lines 302-303)
2.22	2	L358 – although according to table 2 delta AIC for this model was zero (so no model improvement)... it's hard to figure out what criteria the authors are using in each instance (p value, confidence interval, AIC); this should be made clearer and ensure consistency.	P-values have been removed throughout the paper. Delta AIC has been changed to reflect the difference between the null model and each other model (rather than the difference between each model and the model with the lowest AIC) See response to comment 1.12/1.13
2.23	2	L403 – It would be useful to put these results in a wider context by commenting on how trends forest vertebrate populations compare to overall LPI, e.g. are they faring better/worse/similar?	We have added text into the first paragraph of the discussion to put the results in context of trends in terrestrial species globally and of freshwater species (lines 401-406)
2.24	2	L411 – I suggest rephrasing to '...whether the latest upturn in the FSI is a sign of a significant, longer-term increase in the abundance of...'	Incorporated the suggested phrase to Lines 412-414
2.25	2	L416 – and from declining to stable?	Added 'stable' to the current sentence: "...improvement in species trends from declining to increasing or stable" (line 419)
2.26	2	L418 – In what way is the data turnover taking place? E.g. x declining species 'leaving' the dataset vs x increasing species 'entering'...	Added an example of the turnover in data to illustrate this issue: 'This turnover in data is observed in our data set: for example between 2000 and 2002, data for 12 declining and 4 increasing species ended at the same time as data for 10 increasing and 4 declining species entered the data set' (lines 421-423)
2.27	2	L458-466 – I suggest including a few sentences about the potential mechanisms driving some of these trends.	This section of the discussion has been removed, as it related to the cross-correlation

			analyses and the lit review, which have been removed.
3.1	Associate Editor	Both reviewers questioned the claim by the authors that forest cover is commonly used as a proxy for forest biodiversity while the relationship between the two has never been tested. This claim needs to be substantiated by references	See response to comments 1.1 and 2.3. We have removed reference to 'proxy indicator' instead using 'biodiversity-relevant indicator', which is substantiated by the three examples (with references) then given: the CBD, SDGs, and FRA
3.2	Associate Editor	It is also important to distinguish between relationship between forest cover and biodiversity and changes in forest cover and biodiversity. Both were tested in this study, but there is no reason why both should show the same trends, e.g. if there is a threshold of % forest cover above which changes in forest cover are not significantly affecting biodiversity, then different outcome might be expected (see comment by reviewer 1 about the importance of exploring the interactions between predictors instead of considering predictors individually).	See response to comment 1.9
3.3	Associate Editor	Another problem picked up by the reviewer 1 is that for the second part of their analyses, the authors have focused only on 71 populations for which significant relationship with forest cover was found. This resulted in exclusion of 104 populations with no correlation with forest cover from the analysis. The reviewer pointed out that it would be also very important to know the characteristics of species for which no relationship with forest cover has been detected and how many of these species are also threatened by the various pressures examined.	Cross correlation analyses//lit review have been removed.
3.4	Associate Editor	In addition, this reviewer has questioned the methodology used for categorizing cross-correlations into positive, negative or uncorrelated with forest cover, which is very much akin to vote-counting. I share this concern. The $2 \cdot \sqrt{n}$ approach needs to be better justified and ideally a more robust methodology used.	Cross correlation analyses//lit review have been removed.

Appendix B

Response to Referees

Number	Comment	Response (line numbers refer to clean manuscript)
1	The authors need to be clear whether they are looking at forest cover from remote sensing data or forest change. The compare forest change with index change, but in the text refer to both forest cover and cover change. Consequently, it is unclear to me what they want the reader to focus on. The matter is confused more by their presenting a correlation between forest cover from Song et al and Hanse for 2010, but not forest change. I see this was requested in the previous set of reviews too (1.4) but was not implemented. It is essential that they present a correlation of forest change between a comparable time period (2000 – 2010) for their study sites using both sets of satellite image data. Without this it is not possible to assess how consistent the measures of change are.	We have changed the correlation analysis to look at tree cover change 2000-2010 as requested (lines 243-249): “In order to examine the agreement between the two tree cover datasets, we calculated tree cover change per population from 2000 to 2010 using values derived from the Song dataset and from the Hansen dataset. We then assessed the correlation between the two sets of tree cover change values for the 685 populations included in the Hansen analyses. The correlation between the two datasets was highly significant but had a low correlation coefficient (Pearson correlation coefficient = 0.171; $p < 0.001$). This is in agreement with other studies that have found discrepancies between tree cover datasets when assessing tree cover change or area (Gross et al., 2017; Sexton et al., 2016).” We have added to the Discussion (lines 418-420): “Further, discrepancies between satellite-derived tree cover datasets in estimates of tree cover change or area indicate the uncertainties associated with tree cover assessments (Sexton et al., 2016; Gross et al., 2017).”

2

Related to this, I disagree with sentence lines 103 to 105 which is a fundamental point of the paper language but not content – you look at change in indices. Forest cover will not correlate with FSI for obvious reasons (100 ha boreal forest will have very different FSI than 100 ha humid tropical forest), but instead it might correlate with change. This brings me to another point which is would we expect change in 10 ha of one to result in change of similar magnitude in biodiversity in another. Fitting interaction terms to the models of FSI and satellite data would help examine this effect.

We agree with the reviewer that we need to make the language clearer here and that it's not the magnitude of the forest cover that we were explicitly testing, but how much it has changed. This allows us to compare large areas of forest that may support large populations, alongside small areas that may support small populations. Changes in these two may be similar, while reflecting large changes in actual areas (or animals). The FSI is a measure of change rather than an absolute value of abundance so we have made this distinction clearer.

We changed the sentence in Ln 106 to refer to tree cover change.

We also made this reference to change clear in the Discussion (Ln 418 and 453)

On the second point, we would not necessarily expect a linear relationship between habitat change and biodiversity change. This has been raised as a caveat for using remote sensing data for predicting changes in species population trends.

Our approach of modelling abundance change in response to change in forest cover was to capture that while large areas may support larger populations, and smaller areas support smaller - proportional changes should be similar. Particular areas may have differential responses (which would in part be captured by the 'location' random effect), but any consistent relationship between changes in cover and changes in abundance should be captured.

3	The authors need to present a better explanation of why they chose the correlates that they did (line 262) as currently none given. Otherwise it appears an eclectic pick and mix selection. Was the selection based on previous studies or assessment of level 2 or 3 IUCN threats?	We have now specified that correlates were selected a) if they were encoded in the LPD, or b) through literature review (lines 255-256).
4	In their summation, the authors correctly and importantly state the need to have field measurements of biodiversity rather than relying on remote sensing alone. They are not the first to state this, and perhaps some recognition that this is oft stated would balance the conclusions a bit.	We agree, and have added “as has been advocated elsewhere” to this statement, with three supporting references (line 449).
5	Title – It’s a personal thing, but I do not like titles with colons as one part of the title can almost always be deleted. I suggest deleting below the canopy here.	We have discussed and would prefer to keep the title as it is, as the request was based on the reviewer’s personal preference.
6	Ln 20 – you look at the correlation with forest change first – change wording to note this.	Changed “drivers” to “correlates” (line 20)
7	Ln22 – It is not effects of satellite derived tree cover trends, it is a correlation between SPI and tree cover from sat data that you look at. There is a very important difference. Ln 25 is okay.	Reworded to “We analysed the relationships between the average rate of change of forest vertebrate populations and satellite-derived tree cover trends, as well as other pressures.” (lines 21-22)

8	Ln 27 Monitoring alone will not allow forest species to recover. Much more must be done. Change wording here to reflect fact that monitoring allows us to track recovery and implement adaptive management or wording along those lines	This has been changed to “For forest biodiversity to recover, conservation management needs to be informed by monitoring all threats to vertebrates, including those below the canopy.” (lines 26-28)
9	Ln35 – Not really a deadline, rather review date.	We have replaced ‘2020 deadline for the ABTs’ with ‘2020 expiration of the ABTs’. (line 35)
10	Ln 35 – delete legally established unless in FAO wording – it would be interesting to see an illegally established protected area.	“Legally established” is the terminology used by the FAO, and therefore has been kept in (see page 29 of “Global Forest Resources Assessment 2015: How are the world’s forests changing?”)
11	Ln 86 – use °C	Changed to °C (line 86)
12	Ln 90 – delete ‘Recent’ as it is a time limited term in this case	Deleted “recent” (line 88)
13	Ln 102 – derived from rather than using as you use correlation to assess the relationship	Changed “using” to “derived from” (line 101)
14	Ln 106 – 109. Point 1 is not a hypothesis as currently worded	Deleted “given the decline in area of natural forest over time (Keenan et al. 2015) and the link between habitat loss and biodiversity loss (Brooks et al. 2002)” from the first hypothesis. We moved this sentence to the preceding paragraph to explain our hypothesis that forest vertebrates were declining (as an explanation behind this hypothesis was requested in earlier reviewer comments) (lines 98-100).
15	Ln 119 – Data are, not data is. Use if ‘is’ jars when reading. Also Ln 179.	“Is” changed to “are” (line 114), and “was” changed to “were” (twice) (line 171-172)

16	Ln 180 – six, not 6	Changed “6” to “six” (line 172)
17	Ln 184 – what is a biologically useful resolution? Is it a resolution that is appropriate to the size of your study sites from which the population data came? That is probably a more important reason.	We have changed the wording from referring to a “biologically useful resolution” to a “resolution that is sensitive to the local changes that are likely to be relevant to populations” (line 176-177)
18	Ln 218 – 222 – this section appears to be out of place and should be in the additional correlates section.	This has been moved to Supplementary Materials (SM 3).
19	LN 253 – 255 – this is a massive reduction in the number of populations used. You drop specialists by 200 for Hansen based on my calculation. Is there a reason for this as to me this drop indicates a serious issue with these data? I think you need to add more of an explanation as to why you go from 268 (ln 137) to 74. It might be I have missed something in the text.	The drop in number of populations used in the analyses is due to the requirement that populations have at least 2 years of data covering at least a 5 year period within the period covered by the tree cover data sets, i.e. 1982-2016 for Song dataset and 2000-2015 for Hansen dataset (lines 221-224). The drop in the number of populations was particularly noticeable for the Hansen analyses because the period over which the Hansen tree cover data was available (2000-2015) was narrower than that of the Song tree cover data (1982-2016). Therefore, fewer populations met the requirement when analysing tree cover change with the Hansen data set. This is one of the downfalls for our analyses of the higher resolution, shorter-term Hansen dataset and is one of the reasons we additionally ran our analyses using the longer-term lower resolution Song dataset (noted in lines 189-192). We have added an additional sentence for clarity (lines 238-241): “ Fewer populations were included in the analyses with the Hansen dataset because the shorter temporal period covered by the Hansen dataset (2000-

		2015) meant fewer populations had data overlapping that period, compared to the longer-term Song dataset (1982-2016).”
20	Ln 257 – 260 – This is a measure of correlation in area, but you are looking at change in the analysis. Would it be possible to look at the correlation between change 2000 – 2010 in both datasets as a more relevant measure of agreement?	We have changed the correlation analysis to look at tree cover change 2000-2010 as requested (lines 243-249): “In order to examine the agreement between the two tree cover datasets, we calculated tree cover change per population from 2000 to 2010 using values derived from the Song dataset and from the Hansen dataset. We then assessed the correlation between the two sets of tree cover change values for the 685 populations included in the Hansen analyses. The correlation between the two datasets was highly significant but had a low correlation coefficient (Pearson correlation coefficient = 0.171; $p < 0.001$). This is in agreement with other studies that have found discrepancies between tree cover datasets when assessing tree cover change or area (Gross et al., 2017; Sexton et al., 2016).” We have added to the Discussion (lines 418-420): “Further, discrepancies between satellite-derived tree cover datasets in estimates of tree cover change or area indicate the uncertainties associated with tree cover assessments (Sexton et al., 2016; Gross et al., 2017).”

21	Ln 311 – delete ‘wished’ and reword, but otherwise good to see the inclusion of this analysis here.	Changed “wished to determine” to “investigated” (line 296)
22	Ln 342 – Are these SE or 95% Cis around parameter estimates? The non-overlapping nature of the bars in Figure 2 is interesting and worth mentioning. It isn’t mentioned anywhere in the manuscript, but is the massive decline in herptiles due to Chytrid?	These are SEs shown on Figure 2. We have added the missing information to the figure legend. On reviewing this figure we also noticed that part of the y axis label was incorrect and have also amended that. The lack of overlap between error bars in both the taxonomic plot and the one for biomes has been added to the results (Ln 322 and 329) and in the discussion (Ln 365 onwards). We were not able to quantify the effects of different threats on forest populations but we suspect that Chytrid could be one of the main drivers in the amphibian declines, given the threat information we have available. We have added a sentence to the discussion on this to highlight the likely importance of this threat (Ln 368-370)
23	Figure 3 – given the notable differences in trends in figure 2 b and c, I think it is useful to use different symbols for either realm or biome in this figure (as long as it is not a mess) as I am curious to see what this looks like.	We have amended Fig 3 using different symbols to reflect biome (tropical or temperate). We chose the biome option over realm because realm was too messy. We would be interested to know if the reviewer thinks this is now messy and can provide the previous figure without the symbols if preferable.
24	Ln 387 – on average halved or just halved?	The index value is the average of the species trends, so we have kept the reference to ‘on average’ in line 358 but moved it to further along in the sentence to make it clearer: ‘..the global abundance of forest specialists more

		than halved, on average, from 1970 to 2014.'
25	Ln 431- The use of satellite remote sensing data for monitoring populations is often caveated with this very point. While there is no need to undertake a thorough review of the literature it might be worth stressing this point.	Changes to the first two sentences of this paragraph have been made to emphasise the point further (lines 407-411). The point is further picked up (with new references) in the last paragraph of the discussion (line 449).
26	Ln 436 – So what is the implication of this for updating red list categories using remote sensing data (e.g. Tracewski et al 2016)?	This is a valuable point and we have added this, including the suggested reference, to our discussion (Ln 410 - 415). Our results do support some of the caveats noted when using remote-sensing data for extinction risk assessments, for example the assumption of a linear relationship between habitat change and population trends (Tracewski et al 2016) and the influence of threats other than habitat loss (Santini et al 2019).